# Comparative Genomic Analysis Reveals Potential Pathogenicity and Slow-Growth Characteristics of Genus *Brevundimonas* and Description of *Brevundimonas pishanensis* sp. nov.

Zhenzhou Huang,[a,b] Keyi Yu,[a,b] Yue Xiao,[a,b] Yonglu Wang,[c] Di Xiao,[a] Duochun Wang[a,b]

aNational Institute for Communicable Disease Control and Prevention, Chinese Center for Disease Control and Prevention (China CDC), State Key Laboratory of Infectious Disease Prevention and Control, Beijing, People's Republic of China
bCenter for Human Pathogenic Culture Collection, China CDC, Beijing, People's Republic of China
cMa'anshan Center for Disease Control and Prevention, Ma'anshan, Anhui Province, People's Republic of China

**ABSTRACT** The genus *Brevundimonas* consists of Gram-negative bacteria widely distributed in environment and can cause human infections. However, the genomic characteristics and pathogenicity of *Brevundimonas* remain poorly studied. Here, the whole-genome features of 24 *Brevundimonas* type strains were described. *Brevundimonas* spp. had relatively small genomes (3.13 ± 0.29 Mb) within the family *Caulobacteraceae* but high G+C contents (67.01 ± 2.19 mol%). Two-dimensional hierarchical clustering divided those genomes into 5 major clades, in which clades II and V contained nine and five species, respectively. Interestingly, phylogenetic analysis showed a one-to-one match between core and accessory genomes, which suggested coevolution of species within the genus *Brevundimonas*. The unique genes were annotated to biological functions like catalytic activity, signaling and cellular processes, multisubstance metabolism, etc. The majority of *Brevundimonas* spp. harbored virulence-associated genes *icl*, *tufA*, *kdsA*, *htpB*, and *acpXL*, which encoded isocitrate lyase, elongation factor, 2-dehydro-3-deoxyphosphooctonate aldolase, heat shock protein, and acyl carrier protein, respectively. In addition, genomic islands (GIs) and phages/prophages were identified within the *Brevundimonas* genus. Importantly, a novel *Brevundimonas* species was identified from the feces of a patient (suffering from diarrhea) by the analyses of biochemical characteristics, phylogenetic tree of 16S rRNA gene, multilocus sequence analysis (MLSA) sequences, and genomic data. The name *Brevundimonas pishanensis* sp. nov. was proposed, with type strain CHPC 1.3453 (= GDMCC 1.2503[T] = KCTC 82824[T]). *Brevundimonas* spp. also showed obvious slow growth compared with that of *Escherichia coli*. Our study reveals insights into genomic characteristics and potential virulence-associated genes of *Brevundimonas* spp., and provides a basis for further intensive study of the pathogenicity of *Brevundimonas*.

**IMPORTANCE** *Brevundimonas* spp., a group of bacteria from the family *Caulobacteraceae*, is associated with nosocomial infections, deserve widespread attention. Our study elucidated genes potentially associated with the pathogenicity of the *Brevundimonas* genus. We also described some new characteristics of *Brevundimonas* spp., such as small chromosome size, high G+C content, and slow-growth phenotypes, which made the *Brevundimonas* genus a good model organism for in-depth studies of growth rate traits. Apart from the comparative analysis of the genomic features of the *Brevundimonas* genus, we also reported a novel *Brevundimonas* species, *Brevundimonas pishanensis*, from the feces of a patient with diarrhea. Our study promotes the understanding of the pathogenicity characteristics of *Brevundimonas* species bacteria.

**KEYWORDS** *Brevundimonas*, comparative genomics, new species proposed, pathogenicity, slow growth

Address correspondence to Duochun Wang, wangduochun@icdc.cn.

The authors declare no conflict of interest.

Gram-negative, nonfermenting bacteria have raised increasing concern in clinical practice, since they are one of the most common causes of nosocomial infection. Among these, some are well-known opportunistic pathogens associated with hospital-acquired infections, for example, *Pseudomonas aeruginosa* (1), *Acinetobacter baumannii* (2, 3), and *Enterococcus faecium* (4). *Brevundimonas* spp. are relatively less known, but they are also opportunistic human pathogens potentially related to hospital infections.

The genus *Brevundimonas* belongs to family *Caulobacteraceae*, first described by Segers et al. (5) in 1994, and comprises a group of bacteria that share basic microbiological characteristics, like Gram-negative, motile, rods 0.5 $\mu$m by ~1 to 4 $\mu$m, nonfermenting, oxidase-positive, and aerobic or facultative anaerobic. There are currently 35 species within this genus (https://www.bacterio.net/genus/brevundimonas). A profusion of new members can be isolated from diverse sources, such as soil (6–9), lake or sea sediment (10–12), activated sludge (13), aquatic water (14, 15), and human hosts (16–19). In humans, *Brevundimonas* spp. were isolated from eye, blood, urine, skin wound, and the central nervous system and were found in the lung sputum of cystic fibrosis patients (20). The most common and clinically relevant pathogenic species in the *Brevundimonas* genus are *Brevundimonas vesicularis* (related to approximately 70% cases reported) and *Brevundimonas diminuta* (related to more than 20% cases reported). In addition, cases of infection with other *Brevundimonas* bacteria, for example *Brevundimonas vancanneytii*, were also reported (21). In the disease spectrum of the nonmixed infections caused by *Brevundimonas* spp., the common clinical manifestations were ranked by the following, bacteremia, septicemia/sepsis, pneumonia/pleuritis, endocarditis, keratitis, and urinary tract infection (20). However, no case of *Brevundimonas* spp. isolated from the stool samples of patients suffering from diarrhea has been reported so far.

At present, there are few literature reports on diseases caused by *Brevundimonas* spp. (16–19), which makes the pathogenicity of *Brevundimonas* easily overlooked. Studies on the genetic characteristics of the *Brevundimonas* genomes are fewer. Some articles have underreported fragmented introduction to the whole-genome sequencing (WGS) of a certain *Brevundimonas* species. Comparative genomics of the *Brevundimonas* genus remained poorly studied. Few articles have checked the virulence genes and drug resistance genes in detail within the genus *Brevundimonas*. The lack of case reports worldwide may account for the lack of deep understanding of this genus. However, it is warranted to understand their pathogenicity and epidemiology, since the opportunistic infections caused by *Brevundimonas* spp. are under potential epidemic proportions.

In the present study, we explored the evolutionary features of *Brevundimonas* spp. based on WGS and also performed a detailed comparative genomics analysis at the genus level. Meanwhile, we reported a novel species, *Brevundimonas pishanensis*, from the stool sample of a patient with diarrhea. Polyphasic approaches, involving biochemical analysis, 16S rRNA gene phylogenetic analysis, multilocus sequence analysis (MLSA), and whole-genome analysis, were performed to determine the taxonomic position of *Brevundimonas pishanensis* and to systematically describe its phenotypic and genetic characteristics.

## RESULTS

**General features of *Brevundimonas* species genomes.** The average genome size of the 24 type stains (or representative strains) of *Brevundimonas* spp. was 3.13 $\pm$ 0.29 Mb, ranging from 2.54 Mb (*B. halotolerans* MCS24[T]) to 3.75 Mb (*B. subvibrioides* ATCC 15264[T]). This is relatively smaller than that of other genera among the family *Caulobacteraceae*, whose average genome size was 4.49 Mb (Table S3) (Student's *t* test, $P < 0.001$). The average G+C content was 67.01 $\pm$ 2.19 mol%, ranging from 60.8 mol% (*B. terrae* DSM 17329[T]) to 70.4 mol% (*B. viscosa* CGMCC 1.10683[T]), while it was 65.6 mol% in the family *Caulobacteraceae* (Table S3). The genomes among the *Brevundimonas* genus were found to consist of 3,131 protein-encoding genes on average. The draft genome size of the novel strain CHPC 1.3453[T] was 2,916,570 bp, with 17 assembled scaffolds and $N_{50}$ length of 474,365 bp. The G+C content of this strain was 61.6 mol%, and there were approximately

2,800 coding sequences (CDSs) and 47 tRNAs predicted in this genome. The annotation features in subsystems are shown in Fig. S1.

**Evolutionary conservation and diversity of *Brevundimonas*.** We initially studied the evolutionary status and genetic relationship of the genus *Brevundimonas* in the entire *Caulobacteraceae* family. The NJ tree based on 346 single-copy homologous gene sequences demonstrated that the *Brevundimonas* species can be clearly distinguished from other genera (Fig. S2, Table S3). In an attempt to evaluate the conservation between different species within the *Brevundimonas* genus, we analyzed the pairwise homologous gene rate (PHGR) in the representative genomes of any two species. The violin plot (Fig. 1A) showed that the average PHGR for each single species varied from 74.96% (*B. bullata*) to 48.37% (*B. abyssalis*). The PHGR of the novel strain CHPC 1.3453$^T$ ranged from 45.22% (*B. denitrificans*) to 89.52% (*B. bullata*), compared with other 23 species within genus *Brevundimonas*.

From the taxonomic perspective, the average nucleotide identity (ANI) analysis supported the species assignment for the genus of *Brevundimonas*. The overall ANI values between any two representative genomes were under the classical boundary of 95% to 96% (22, 23) for an independent species or subspecies (Fig. 1B), except for two groups, i.e., *B. diminuta* ATCC 11568$^T$ - *B. vancanneytii* NCTC 9239 and *B. abyssalis* TAR-001$^T$ - *B. denitrificans* TAR-002$^T$. It suggested that each group belonged to synonyms.

We analyzed the pan genome diversity of *Brevundimonas* spp. by using the Roary pipeline (Fig. 1C). A total of 17,194 genes were defined in 24 genomes, and 762 of these genes were shared by all species, which formed a set of the so-called hard core genes. Also, about 300 soft core genes were harbored by 95% to 99% of the genomes. Hard core genes together with soft core genes accounted for approximately 1/16 of the pan genes. According to the gene presence or absence profile similarity of 25.6% as the cutoff, the hierarchical clustering dendrogram of pan genome indicated that *Brevundimonas* spp. can be divided into 5 major clades (Fig. 1C). Clade II contained the largest number of *Brevundimonas* species (*n* = 9), followed by clade V (*n* = 5), where the novel strain CHPC 1.3453$^T$ was located. The conserved gene and pan gene dilution curve demonstrated that the genomes of *Brevundimonas* spp. were open and compatible (Fig. 1D). As the number of genomes increased, the number of conserved genes tended to be stable, while the number of pan genes kept increasing.

**Genetic structures of closely related members of *Brevundimonas* spp.** To further reveal the genomic variation within the genus *Brevundimonas*, we constructed two phylogenetic trees based on the set of core and accessory genomes, respectively (Fig. 2A). Like the two-dimensional hierarchical clustering of pan genome, the core genome phylogeny also reflected the tendency toward clades I to V and formed different evolutionary branches or clusters. Surprisingly, the phylogenetic tree based on core genome had a topology highly similar to that based on accessory genome, and the *Brevundimonas* spp. within the 5 clades also formed a one-to-one mapping parallelism. This suggested that *Brevundimonas* spp. may have a relationship of interactive coevolution.

When the genome of the novel strain CHPC 1.3453$^T$ was compared to the genomes of *Brevundimonas* spp. in clade V, approximately hundreds of localized colinear blocks (LCBs) were estimated by MAUVE comparison (Fig. 2B, Fig. S3). *B. terrae* was the species most closely related to the novel strain CHPC 1.3453$^T$ in terms of genome structure, with the smallest number of LCBs at 138, while *B. bullata* was the most distantly related one, with the largest number of LCBs at 326. Comparing the genome of CHPC 1.3453$^T$ to that of *B. terrae*, 472,805 single nucleotide polymorphisms (SNPs) were calculated. The Venn diagram (Fig. 2C) showed the shared and unique genes found in six *Brevundimonas* genomes, and 432, 1,278, 1,205, 394, 1,047, and 665 unique genes were identified in each genome, respectively. These unique genes were annotated by GO, COG, and KEGG, and many genes (except the hypothetical protein) were related to biological functions such as catalytic activity, signaling and cellular processes, multisubstance metabolism (like amino acid, nucleotide, carbohydrate), etc. (Fig. S4). Therefore, it was reasonable to speculate that *Brevundimonas* species genomes might undergo many sorts of rearrangements, since

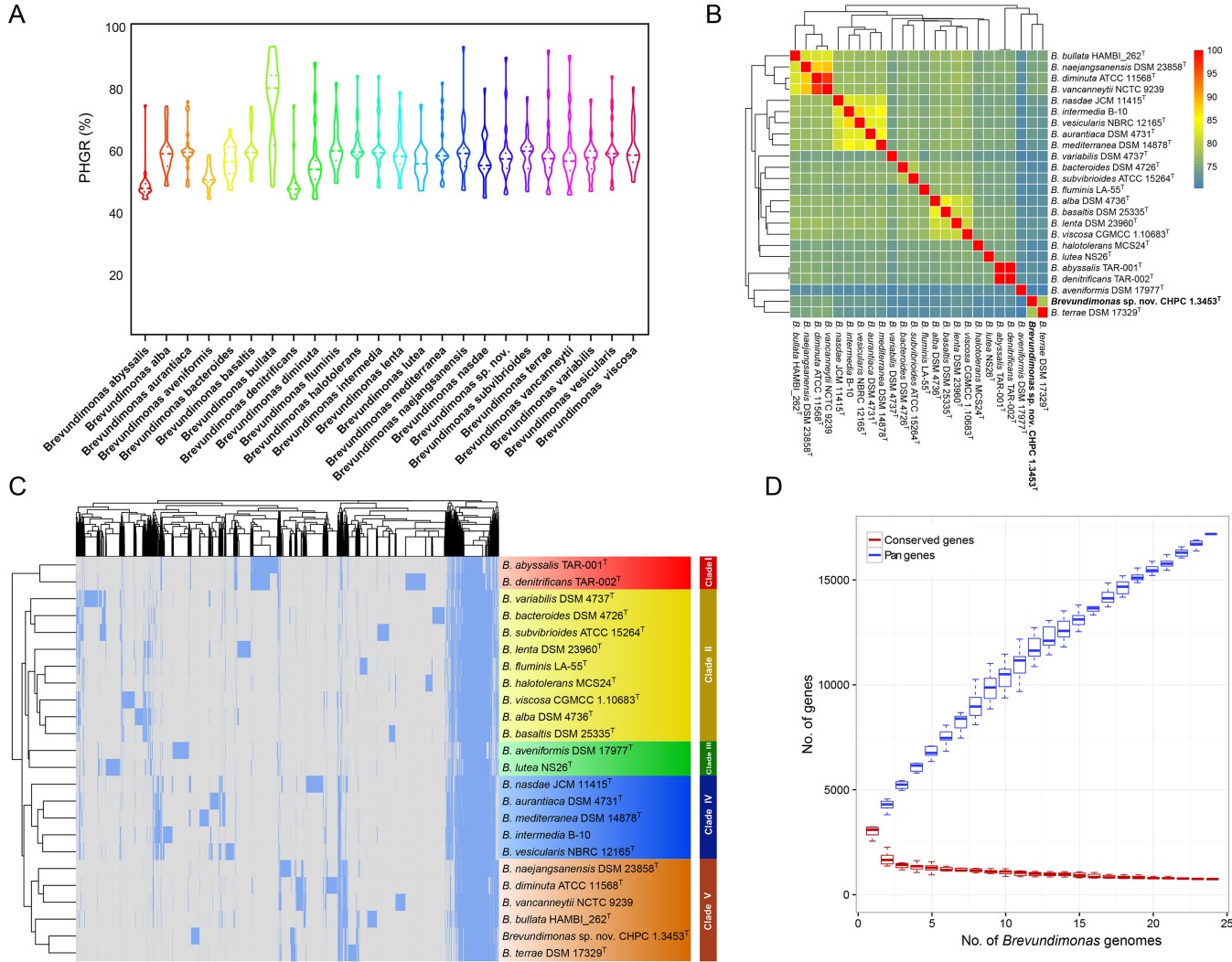

**FIG 1** General genomic characteristics and diversity of *Brevundimonas* genus. (A) The pairwise homologous gene rate (PHGR) of 24 *Brevundimonas* species genomes. The bottom and top of the violin plots indicate 0.25 and 0.75 quantiles, respectively. The dashes represent the median of each species in rainbow colors, and the bandwidth represents the density distribution. (B) Average nucleotide identity (ANI) among *Brevundimonas* members based on their whole-genome sequences. The colors from blue to red indicate a gradual increase in ANI values. (C) Pan genome diversity of *Brevundimonas* spp. for which genomes are available. The two-dimensional hierarchical clustering of species and genes is based on the presence (blue) or absence (gray) of genes. The red, yellow, green, blue, and brown on the right refer to the evolution of clades I to V, respectively. (D) Conserved gene and pan gene dilution curves of *Brevundimonas* genomes. The red boxplot represents the number of conserved genes, and the blue represents the number of pan genes.

massive LCBs indicate events relating to changes in genome structure, including insertions, deletions, inversions, and translocations.

**Distribution of potential virulence-associated genes, antimicrobial resistance genes, and mobile genetic elements.** Through comparison of the *Brevundimonas* genomes with the Virulence Factor Database (VFDB), we found that *Brevundimonas* spp. shared virulence-associated genes homologous to 20 species from different genera, with *Brucella melitensis*, *Francisella tularensis*, and *Legionella pneumophila* the top three species sharing the largest number of virulence-associated genes (Fig. 3A). The virulence-associated genes in the novel strain CHPC 1.3453[T] were sourced from the above three species as well as *Mycobacterium tuberculosis* and *Helicobacter pylori*. The majority of *Brevundimonas* spp. (including CHPC 1.3453[T]) harbored the genes *icl* (VFG001381), *tufA* (VFG046465), *kdsA* (VFG011414), *htpB* (VFG001855), and *acpXL* (VFG011430), which encoded isocitrate lyase, elongation factor Tu, 2-dehydro-3-deoxyphosphooctonate aldolase, heat shock protein HtpB, and acyl carrier protein, respectively. In particular, the genes *kdsA* and *acpXL* were detected in 95.8% (*n* = 23/24) of *Brevundimonas* species, which were related to LPS with the function of adhesion and endotoxin (Fig. 3B).

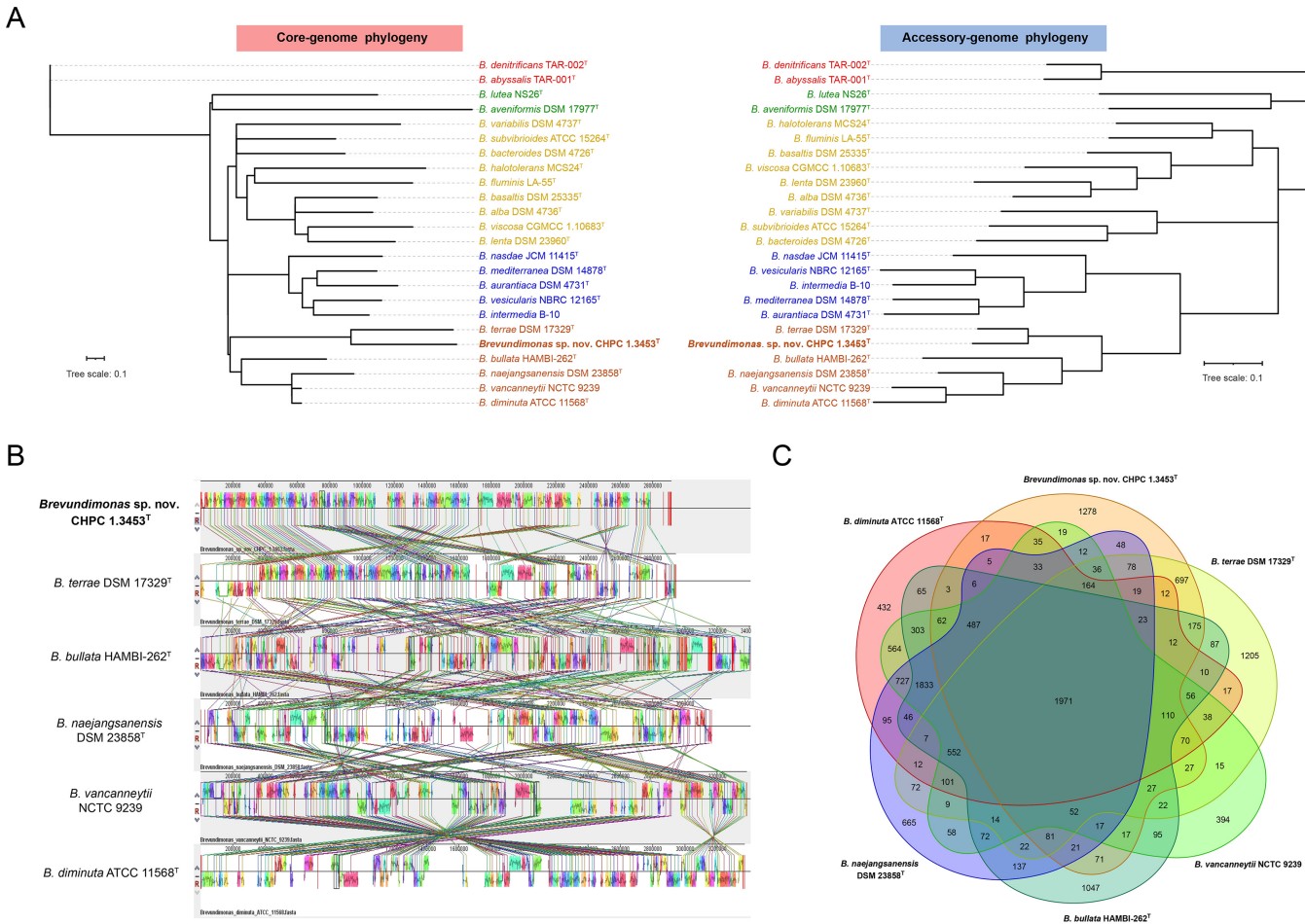

**FIG 2** The evolutionary relationship of the novel *Brevundimonas* strain CHPC 1.3453ᵀ. (A) The core and accessory genome phylogenies of 24 *Brevundimonas* species strains. The color of each species corresponds to the color of clades I to V in Fig. 1C. (B) The MAUVE comparison of CHPC 1.3453ᵀ and closely related members in clade V of pan genome clustering in Fig. 1C. (C) The Venn diagram of the shared and unique genes found in the novel strain CHPC 1.3453ᵀ and other closely related members in clade V.

The majority of the *Brevundimonas* species (87.5%, *n* = 21/24) were detected to carry one or more drug resistance genes (Table S4). *B. diminuta* and *B. naejangsanensis* harbored the most antimicrobial resistance genes, while other species, including the novel strain CHPC 1.3453ᵀ, harbored only one gene, namely, an antibiotic efflux pump-related gene *adeF*. Apart from *adeF*, *B. diminuta* contained a tetracycline resistance gene, *tet*(C), while *B. naejangsanensis* contained a sulfonamide resistance gene, *sul2*, as well as two tetracycline resistance genes, *tet*(D) and *tet*(G) (Table S4).

We also analyzed the distribution of mobile genetic elements (MGEs; for example, genomic islands [GIs], plasmids, CRISPR-cas, phages, and prophages) in 24 *Brevundimonas* species. The GIs were identified by three algorithms of IslandPath-DIMOB, SIGI-HMM, and IslandPick, and the number of GIs ranged from 8 (*B. aveniformis* DSM 17977ᵀ) to 79 (*B. vancanneytii* NCTC 9239). Detailed information of the cross-species or the cross-genus GIs is listed in Table S5 and Table S6, respectively. Notably, the GIs from the novel strain CHPC 1.3453ᵀ were highly homologous to GIs from *B. subvibrioides* ATCC 15264 (accession number CP002102.1) with 100% coverage and 100% identity. Meanwhile, 3 intact and 12 hypothetical phage/prophage sequences were detected by PHASTER (Table S7). Strain CHPC 1.3453ᵀ was found to contain a complete phage sequence, which was genetically homologous to the sequence of PHAGE_Ralsto_RsoM1USA (accession number NC_049432). However, no plasmid replicon or CRISPR-cas-related sequence was detected in the representative genomes of *Brevundimonas* genus.

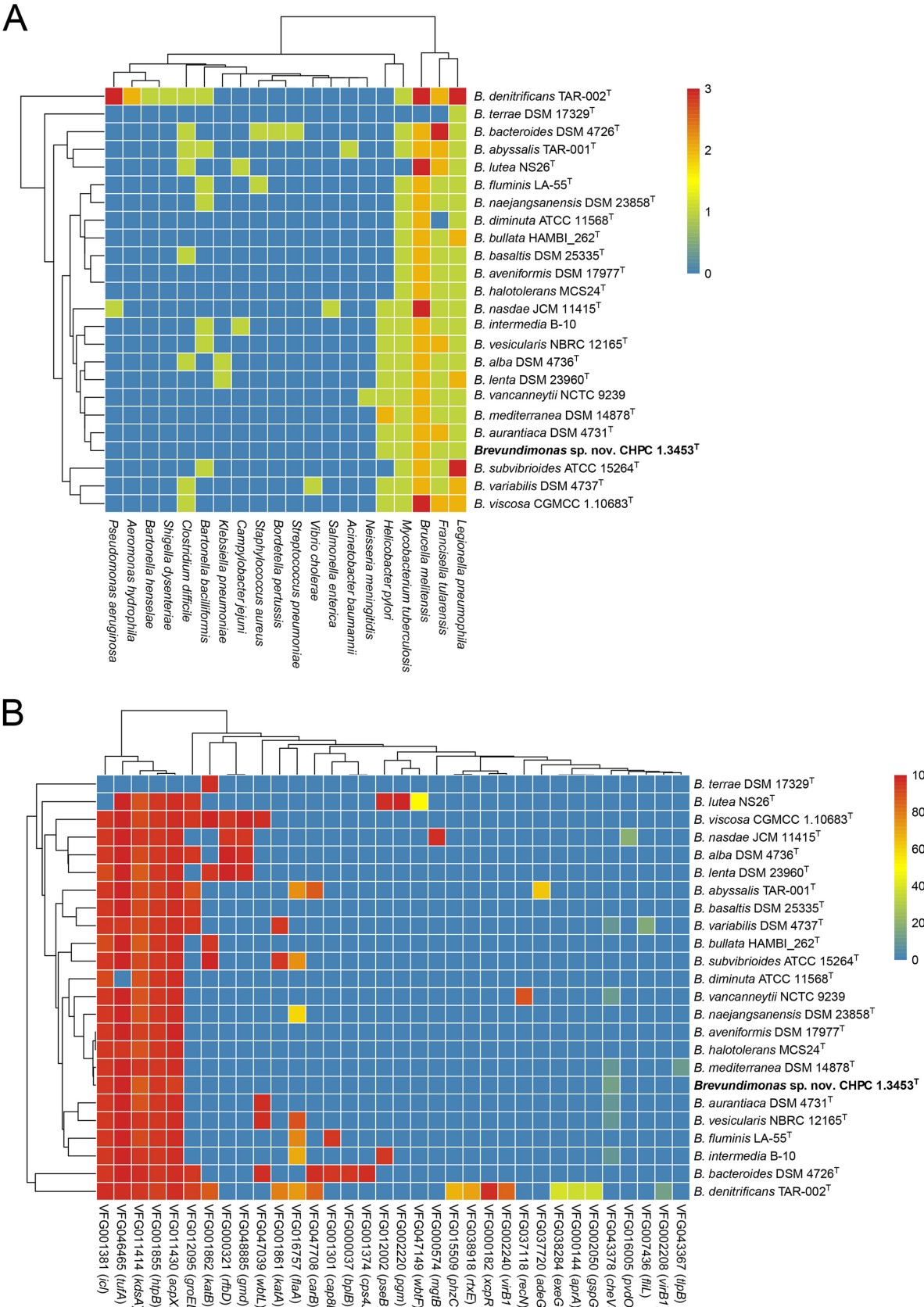

**FIG 3** The distribution of virulence-associated genes of *Brevundimonas* genomes. (A) Hierarchical clustering heatmap of 24 *Brevundimonas* spp. with respect to the virulence gene sources. The colors from blue to red represent the correlation frequency of virulence gene homology. (B) Hierarchical clustering heatmap of virulence-associated genes in 24 *Brevundimonas* spp. The colors represent the sequences' BLAST coverage values of virulence-associated genes.

**Strain CHPC 1.3453$^T$ was identified as a novel species of the genus *Brevundimonas*.** The 16S rRNA gene sequence (1,418 bp) of strain CHPC 1.3453$^T$ was aligned against the corresponding sequences of 33 known *Brevundimonas* species (Fig. 4A). Strain CHPC 1.3453$^T$ was demonstrated to share the highest similarity (98.53%) to *B. terrae* KSL-145$^T$ (DQ335215), followed by *B. diminuta* ATCC 11568$^T$ (GL883089, 97.11%) and *B. faecalis* CS20.3$^T$ (FR775448, 95.34%). The similarity values between the strain CHPC 1.3453$^T$ and some *Brevundimonas* spp. were slightly above 97%, the threshold commonly suggested for species delineation (24). Phylogenetic analysis based on 16S rRNA gene sequences revealed that strain CHPC 1.3453$^T$ belonged to the genus *Brevundimonas*. MLSA method of 5 housekeeping genes (concatenated sequence in the order *gyrB-ppsA-recN-rpoC-rpoD*) was established for the determination of a more refined population structure among *Brevundimonas* species. Strain CHPC 1.3453$^T$ formed an independent and robust interspecies phylogenetic branch and was closely clustered with *B. terrae*. Notably, strain CHPC 1.3453$^T$ shared the highest sequence similarities (83.64%) to *B. terrae* DSM 17329$^T$ in terms of MLSA sequence similarities (Fig. 4B), which was consistent with the results of 16S rRNA gene sequences analysis.

We further calculated the digital DNA-DNA hybridization (dDDH) and ANI values for the strain CHPC 1.3453$^T$ and the closely related representative strains of the genus *Brevundimonas* (Table 1). As expected, strain CHPC 1.3453$^T$ shared 21.70% similarity with *B. terrae* DSM 17329$^T$ in terms of dDDH. In the view of Auch et al. (25), the classical species threshold for dDDH values was 70%, clearly interpreting the differentiation between strain CHPC 1.3453$^T$ and its closest relatives. Consistently, ANI values between strain CHPC 1.3453$^T$ and other closely related type strains (or representative strains) of *Brevundimonas* species ranged from 74.8% to 78.2% (Fig. 1B), lower than the 95% to 96% cutoff value for defining species threshold (22).

Cells of strain CHPC 1.3453$^T$ were motile by a single polar flagellum. The strain formed circular, orange-yellow colonies with a diameter of 0.5 to 1.5 mm on LB agar and gray colonies on blood agar at 35℃ for 18 to 72 h, with translucent texture, round shape, and plump appearance (Fig. 5A and B). Strain CHPC 1.3453$^T$ was small, short, and rod-shaped under high-resolution transmission electron microscopy (Fig. 5C). Cell size was 0.4 $\mu$m in width and 1.2 to 4 $\mu$m in length. Phenotypic results showed that strain CHPC 1.3453$^T$ was Gram-negative and oxidase-positive, it can grow on LB medium, blood medium, BHI medium, or R2A agar, it was able to grow under a wide range of pH values (pH 6 to 10, optimum pH 8) and NaCl tolerance concentrations (0 to 3%, wt/vol), it can grow at a temperature range of 15 to 42℃ (it did not grow at 4℃), the optimum growth temperature was recommended to be 30 to 37℃, and it was facultative aerobic. The strain was susceptible to amikacin (MIC ≤ 8 $\mu$g/mL, the same below), amoxicillin-clavulanate (≤8/4 $\mu$g/mL), ampicillin-sulbactam (≤4/2 $\mu$g/mL), cefepime (8 $\mu$g/mL), cefoperazone-sulbactam (2/8 $\mu$g/mL), cefoxitin (8 $\mu$g/mL), ceftazidime (16 $\mu$g/mL), ceftriaxone (≤1 $\mu$g/mL), cefuroxime (≤4 $\mu$g/mL), chloramphenicol (≤4 $\mu$g/mL), ciprofloxacin (2 $\mu$g/mL), colistin (2 $\mu$g/mL), ertapenem (1 $\mu$g/mL), fosfomycin (32 $\mu$g/mL), gentamicin (≤2 $\mu$g/mL), imipenem (1 $\mu$g/mL), levofloxacin (≤1 $\mu$g/mL), meropenem (0.5 $\mu$g/mL), minocycline (≤1 $\mu$g/mL), moxifloxacin (1 $\mu$g/mL), norfloxacin (4 $\mu$g/mL), piperacillin-tazobactam (≤4/4 $\mu$g/mL), tetracycline (≤2 $\mu$g/mL), tigecycline (≤1 $\mu$g/mL), and tobramycin (≤2 $\mu$g/mL) but resistant to aztreonam (>32 $\mu$g/mL), cefazolin (>16 $\mu$g/mL), nitrofurantoin (>64 $\mu$g/mL), and trimethoprim-sulfamethoxazole (>4/76 $\mu$g/mL).

Members of *Brevundimonas* genus grow slowly on ordinary nutrient media (20), like *B. vesicularis* and *B. diminuta* (26). We conducted a growth rate test for strain CHPC 1.3453$^T$, with the type strains of *B. vesicularis* and *B. diminuta* as slow-growth controls and *Escherichia coli* ATCC 25922$^T$ as a fast-growth control (Fig. 5D). The growth rate of strain CHPC 1.3453$^T$ was approximately equivalent to that of *B. diminuta* ATCC 11568$^T$; the bacterial cells colonized rapidly and the growth rate increased until a constant growth rate was achieved in the 12 to 16 h during the exponential phase. Eventually, both strains CHPC 1.3453$^T$ and *B. diminuta* ATCC 11568$^T$ reached the saturation phase after 24 h, and both of them grew much more slowly than *E. coli* ATCC 25922$^T$ and

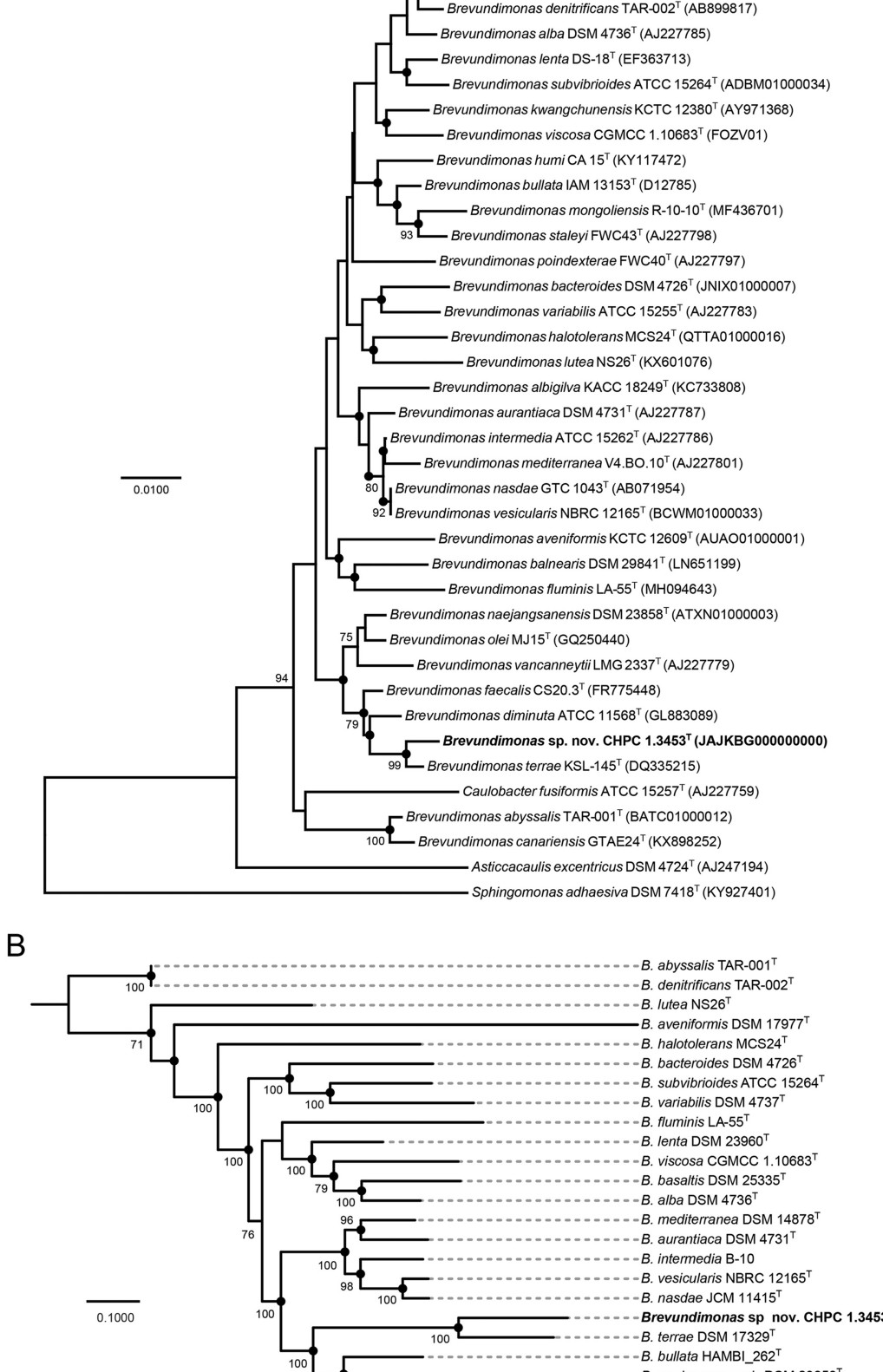

**FIG 4** Phylogenetic trees of *Brevundimonas* species. (A) The neighbor-joining tree based on 16S rRNA gene sequences of the strain CHPC 1.3453ᵀ and other *Brevundimonas* species isolates. Bootstrap values below 70%, based on 1,000 resamplings, are

**TABLE 1** Average nucleotide identity (ANI) and digital DNA–DNA hybridization (dDDH) between strain CHPC 1.3453[T] and closely related representative strains of the genus *Brevundimonas*[a]

| Query genome | Reference genome | Accession no. | dDDH (%) | Model CI | Distance (Mb) | ANI (%) |
|---|---|---|---|---|---|---|
| CHPC 1.3453[T] | *B. bullata* HAMBI_262[T] | QLLC01 | 20.5 | 17.5–22.1% | 0.2 | 74.8 |
| CHPC 1.3453[T] | *B. diminuta* ATCC 11568[T] | ADUI01 | 21.0 | 18.7–23.4% | 0.2 | 75.0 |
| CHPC 1.3453[T] | *B. naejangsanensis* DSM 23858[T] | ATXN01 | 20.4 | 17.2–21.8% | 0.2 | 75.0 |
| CHPC 1.3453[T] | *B. terrae* DSM 17329[T] | JAASQT01 | 21.7 | 18.2–22.8% | 0.2 | 78.2 |
| CHPC 1.3453[T] | *B. vancanneytii* NCTC 9239 | CP002102 | 20.7 | 19.5–24.1% | 0.2 | 75.1 |

[a]The dDDH value based on formula 2 was calculated using the GGDC web server; ANI values were estimated using the web-based service ANI calculator (http://www.ezbiocloud.net/tools/ani). Model CI, model confidence interval.

more quickly than *B. vesicularis* NBRC 12165[T]. From the growth curve (Fig. 5D), the time required for each cell division was calculated, that is, the doubling time or generation time. The generation times of *Brevundimonas* genus bacteria (strain CHPC 1.3453[T] = 75 min; *B. diminuta* ATCC 11568[T] = 81 min; *B. vesicularis* NBRC 12165[T] = 135 min) were much longer than that of *E. coli* ATCC 25922[T] (27 min).

To further determine the characteristics of strain CHPC 1.3453[T], biochemical tests were performed. The results showed that $\beta$-glucosidase, D-xylose, inositol, D-melibiose, potassium 2-ketogluconate, and potassium 5-ketogluconate were assimilated, but the following substrates were not assimilated: L-tryptophane, D-glucose, arginine dihydrolase, urease, protease, 4-nitrophenyl-$\beta$-D-galactopyranoside, glucose, L-arabinose, D-mannitol, *N*-acetyl-D-glucosamine, D-maltose, potassium gluconate, capric acid, adipic acid, malic acid, trisodium citrate, phenylacetic acid, glycerol, erythritol, D-arabinose, D-ribose, L-xylose, methyl-$\beta$-D-xylopyranoside, D-galactose, D-fructose, D-mannose, L-sorbose, L-rhamnose, dulcitol, D-sorbitol, methyl-$\alpha$-D-mannopyranoside, methyl-$\alpha$-D-glucopyranoside, *N*-acetylglucosamine, amygdalin, arbutin, esculin ferric citrate, salicin, D-cellobiose, D-lactose (bovine origin), D-saccharose (sucrose), D-trehalose, inulin, D-melezitose, D-raffinose Amidon (starch), glycogen, xylitol, gentiobiose, D-turanose, D-lyxose, D-tagatose, D-fucose, L-fucose, D-arabitol, and L-arabitol. In API ZYM system assay, alkaline phosphatase, esterase (C4), leucine arylamidase, trypsin, acid phosphatase, and naphthol-AS-BI-phosphohydrolase were present, but esterase lipase (C8), lipase (C14), valine arylamidase, cystine arylamidase, $\alpha$-chymotrypsin, $\alpha$-galactosidase, $\beta$-galactosidase, $\beta$-glucuronidase, *N*-acetyl-$\beta$-glucosaminidase, $\alpha$-mannosidase, and $\alpha$-fucosidase were absent. Detailed information about the characteristics of strain CHPC 1.3453[T] and type strains of other closely related species are listed in Table 2.

The above phenotypic, biochemical, phylogenetic, and genomic analyses clearly indicated that strain CHPC 1.3453[T] belongs to a novel species within the genus *Brevundimonas*. We propose the species name *Brevundimonas pishanensis* (pi.shan.en'-sis. fem. adj. *pishanensis*, referring to Pishan County of Hotan Prefecture [*pishan* in Chinese], where the strain was recovered). The type strain is CHPC 1.3453[T] (= GDMCC 1.2503[T] = KCTC 82824[T]), and the DNA G+C content is 61.6 mol%.

## DISCUSSION

In this study, we provide an extensive overview of the diverse genomic features of the genus *Brevundimonas*, with major emphasis on pan genome evolution, genetic structures, and MGE-related characterization (including potential virulence-associated genes). We conducted a comparative analysis for the genomes of the type strains (or representative strains) of 24 *Brevundimonas* species and analyzed the core and accessory

**FIG 4** Legend (Continued)

not shown at branch nodes. Filled black circles at nodes indicate generic branches that are synchronously recovered by using neighbor-joining and maximum-likelihood algorithms. GenBank accession numbers of the 16S rRNA gene sequences are given in parentheses. Three strains, namely, *Caulobacter fusiformis* ATCC 15257[T], *Asticcacaulis excentricus* DSM 4724[T], and *Sphingomonas adhaesiva* DSM 7418[T], respectively, serve as outgroups. The horizonal bar represents 0.01 substitution per nucleotide site. (B) The maximum-likelihood tree based on 5 housekeeping genes concatenated sequences, in the order *gyrB-ppsA-recN-rpoC-rpoD*. Numbers at nodes indicate bootstrap values (percentage of 1,000 replicates) greater than 70%. Filled black circles indicate generic branches that are also recovered by using neighbor-joining and maximum-likelihood algorithms. The horizonal bar represents 0.1 substitution per nucleotide site.

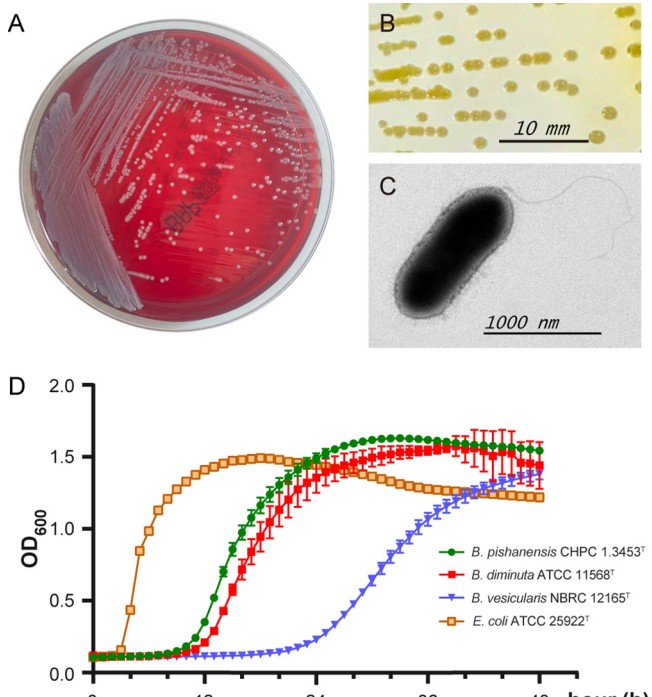

**FIG 5** Culture and morphology characteristics of strain CHPC 1.3453[T]. (A) The growth of this strain after 18 h at 35°C on blood agar medium. (B) The morphology of a single colony after 72-h growth at 35°C on lysogeny broth (LB) medium. The bar represents a unit length of 10 mm. (C) The general morphology of a negatively stained bacterial cell shown by high-resolution transmission electron. The bar represents a unit length of 1,000 nm. (D) Growth curves of strain CHPC 1.3453[T]. Three strains, *B. vesicularis* NBRC 12165 [T], *B. diminuta* ATCC 11568[T], and *E. coli* ATCC 25922[T], were used as controls.

genome sequences based on phylogenetic relationships. We also provided an objective and comprehensive description of the potential virulence and drug resistance genes carried by the *Brevundimonas* spp., which may be a basis for the further exploration of their specific pathogenic mechanisms and drug resistance risks. Furthermore, we reported a novel *Brevundimonas* species based on both phenotypic and genomic analyses. It is the first *Brevundimonas* species recovered from stool of a patient suffering from diarrhea.

The G+C content of bacterial chromosomes varies from 17 mol% to 75 mol% (27), which can serve as a basis for analyzing the relationships between bacterial species or the origin of genes. Meanwhile, the genome size of most bacteria is positively correlated with their G+C contents. The correlation value decreases at shorter genome sizes, where there is a wider spread of G+C values (27). Within the genus *Brevundimonas*, the variation of the genome size was smaller than that of the G+C content (standard deviation of the genome size was smaller than that of the G+C content). *B. subvibrioides* contained the largest genome size, followed by *B. bullata*, *B. mediterranea*, and *B. nasdae*, while *B. viscosa* contained the highest G+C content, followed by *B. fluminis* and *B. lenta*. However, the genus *Brevundimonas* is an exception in terms of the relationship between the genome size and the G+C content. The relatively small average genome size (3.13 ± 0.29 Mb) of the *Brevundimonas* genus was previously presumed to have a range of G+C content extend from 45% to 50% according to the correlation formula (28). In fact, we found that *Brevundimonas* spp. had an average G+C content of up to 67%. It is interesting that *Brevundimonas* spp. have small genome sizes but high G+C contents. One possible reason could be the biases of codon usage (29). Genomic G+C content has been identified as the most important determinant for codon usage (30). Despite synonymous amino acids, unknown selective forces favoring GC-rich codons may influence the G+C content of the genomes of *Brevundimonas* genus.

To gain further clarity on the interspecies relationships, we performed a pan

**TABLE 2** Physiological and biochemical characteristics of strain CHPC 1.3453[T] and closely related species of the genus *Brevundimonas*[a]

| Characteristic | Value for taxon no.: | | | | | | | |
|---|---|---|---|---|---|---|---|---|
| | **1** | **2** | **3** | **4** | **5**[b] | **6**[b] | **7**[b] | **8**[b] |
| Colony pigmentation | Yellow | Yellow | Orange/red | Light yellow | Greyish yellow | NA | Greyish yellow | Whitish yellow |
| Oxidase | + | + | + | + | + | + | + | + |
| Urease | − | − | − | − | + | NA | − | NA |
| Reduction of nitrates to nitrite | − | − | − | − | NA | w or − | NA | − |
| Reduction of nitrates to nitrogen | − | − | − | − | NA | w or − | NA | − |
| | | | | | | | | |
| Assimilation of: | | | | | | | | |
| Glycerol | − | + | − | − | − | − | − | w |
| L-Arabinose | − | w | + | − | w | w | − | + |
| D-Galactose | − | w | + | − | w | w | − | + |
| D-Glucose | − | − | w | − | − | NA | − | + |
| D-Maltose | − | − | + | − | − | − | − | w |
| D-Cellobiose | − | − | + | − | − | − | − | NA |
| Glycogen | − | − | + | w | + | w | − | − |
| Gentiobiose | − | − | − | − | − | − | − | + |
| D-Tagatose | w | w | + | + | NA | NA | − | NA |
| Arginine dihydrolase | − | − | − | − | − | NA | − | NA |
| β-Glucosidase | + | + | + | + | − | NA | NA | NA |
| Protease | − | + | − | − | + | − | NA | − |
| Mannose | − | − | − | − | − | − | − | + |
| N-Acetyl-D-glucosamine | − | − | − | − | − | + | − | + |
| | | | | | | | | |
| API ZYM reactions | | | | | | | | |
| Esterase (C 4) | + | − | + | + | − | − | + | + |
| Esterase lipase (C 8) | − | w | + | w | − | − | − | + |
| Leucine arylamidase | w | + | + | + | w | + | + | + |
| Valine arylamidase | − | − | w | w | w | + | + | w |

[a]Taxa: 1, strain CHPC 1.3453[T]; 2, *B. diminuta* ATCC 11568[T]; 3, *B. vesicularis* NBRC 12165[T]; 4, *B. halotolerans* MCS24[T]; 5, *B. terrae* DSM 17329[T]; 6, *B. bullata* HAMBI_262[T]; 7, *B. naejangsanensis* DSM 23858[T]; 8, *B. vancanneytii* LMG 2337[T]. +, positive; −, negative; w, weakly positive; NA, not available.
[b]Data from Segers et al. (1994), reference 5; Yoon et al. (2006), reference 7; Estrela et al. (2010), reference 21; Kang et al. (2009), reference 9.

genome analysis for the genomes of 24 *Brevundimonas* species. Most of these genomes shared more than 1,000 orthologous groups. From the perspective of evolution, a high-resolution hierarchical clustering phylogram (Fig. 1C) based on the presence or absence of pan genes distinguished the 24 *Brevundimonas* spp. into different clustering branches, which was consistent with the results of ANI values. It should be noticed that *B. pishanensis*, a novel *Brevundimonas* species found in this study, was evolutionarily closest to the *Brevundimonas* spp. in clade V. Interestingly, the phylogenetic analysis of the sequences of core genome and accessory genome showed a one-to-one match, which may suggest coevolution of the core and accessory genomes of species within the genus *Brevundimonas* and equal roles played by the core and accessory genomes over the same period. One possible reason for their coevolution may be horizontal gene transfer. For example, mobile elements have similar outcomes in the variations of core or accessory genome, and the probability of this occurrence is relatively consistent in each species (31). We then did a more refined genome structure comparison for several *Brevundimonas* spp. in clade V. When the genomes were aligned, multiple scattered blocks were identified, suggesting that the genome structures of the novel *B. pishanensis* and other *Brevundimonas* spp. in clade V were genetically distinct, which may be due to a variety of genetic phenomena such as gene rearrangements, inversions, translocations, and insertions.

Opportunistic pathogens are constantly emerging and changing, with little or no apparent virulence, and these bacteria are mostly drug resistant or multidrug resistant (32–35). They are generally nonpathogenic in their indigenous locations, but when the homeostasis in human body is disrupted, they take advantage of the situation and invade the body, causing various diseases. This happened mainly in hospitalized and immunocompromised patients, leading to life-threatening infections with high mortality rates (36). Limited knowledge existed for the pathogenicity of the genus *Brevundimonas*. The majority of the

currently known infections caused by *Brevundimonas* spp. were found to be accompanied by the patients' underlying diseases, for example, urinary tract infections (UTI) (37), bacteremia (16), and empyema (38) in clinical settings. Other co-infection and pseudo-outbreaks associated with *Brevundimonas* spp. have also been rarely reported (37, 39). Despite intensive efforts, treatment strategies currently remained insufficient to eradicate such *Brevundimonas* species infections (20). One aim driving this study was to understand the potential pathogenicity of *Brevundimonas* spp. by predicting the virulence gene profile, particularly the horizontal transfer of virulence genes associated with MGEs within species. We found that most *Brevundimonas* spp., except *B. terrae* DSM 17329$^T$, carried five virulence genes (*icl*, *tufA*, *kdsA*, *htpB*, and *acpXL*). These virulence genes have homologues in species from other genera, such as *Mycobacterium tuberculosis*, and have been confirmed to associate with pathogenicity. For example, isocitrate lyase (ICL), an enzyme essential for the metabolism of fatty acids, plays a pivotal role in one of the fatty acid metabolism mechanisms for *Mycobacterium tuberculosis* persistence (40). Bhusal et al. (41) also demonstrated that ICL2 was critical for bacterial growth and virulence, which was involved mainly in the regulation of carbon fluxes in tricarboxylic acid cycle, glyoxylate shunt, and methylcitrate cycle. Another example is the product encoded by *acpXL* gene, which acts as a donor of C28 fatty acid for lipid A (42) and involves in the transfer of tC28 fatty acid to lipid A precursor. Sharypova et al. proved that *acpXL* mutation indeed blocked the C28 acylation of lipid A (43). AcpXL could be a favorable substance to enhance the biochemical and infection phenotypes of Gram-negative bacteria (44). Our analysis indicated that each *Brevundimonas* species contained a very similar virulence gene profile. However, only *B. vesicularis* and *B. diminuta* were clearly reported to be pathogenic, which accounted for nearly 90% of *Brevundimonas* infections. For drug resistance, *B. vesicularis* was reported to be associated with bacteremia and showed highly variable sensitivity to broad-spectrum antibiotics (45). *Brevundimonas* spp. were also influential factors in the spread of carbapenem resistance (46), which was the characterization of clinical environments. The novel strain, *B. pishanensis*, was found to harbor one antibiotic efflux pump-related gene, *adeF*. No drug-resistant genes was identified in the two *Brevundimonas* species causing human infections, *B. diminuta* and *B. vesicularis*.

The growth rate of bacteria within the genus *Brevundimonas*, including the novel species *B. pishanensis*, was significantly lower than that of *E. coli*, especially in *B. vesicularis*. Bacterial growth is often system-wide linked, including gene expression, growth feedback, and proteome partition (47). It has been demonstrated that the growth rate can serve as a key parameter in assessing the overall bacterial growth (48, 49). Tight regulation of gene expression limits the waste of resources and energy, and specific genes or operons may be regulated by different mechanisms. Using *E. coli* as a paradigm, ppGpp, which was produced by two ppGpp synthetases, RelA and SpoTacting, acted with DksA in a synergistic way and became a major effector in bacterial growth rate regulation (50). Our study suggested that, when the environmental conditions were the same, *Brevundimonas* spp. may have increased ppGpp regulation ability, which then made them have growth rates significantly lower than those of *E. coli*. Also, the nutritional and habitat conditions forced bacteria to exhibit a more diverse growth strategy of nutrient and metabolic versatility. The synergistic effect between ppGpp and DksA above was also a key effector for *E. coli* in the stringent response induced by nutrient starvation (51, 52). Some bacteria were already found to have a greater number and variety of genes encoding regulatory elements, as recently demonstrated by studies that showed that Gram-positive soil bacterium *Bacillus subtilis* and the Gram-negative opportunistic pathogen *Pseudomonas aeruginosa* were able to accumulate a considerable abundance and diversity of transcriptional regulators, two-component systems, and alternative $\sigma$ factors under nutrient- and habitat-adverse conditions (53–55).

Next-generation sequencing (NGS), as an important milestone, has been widely applied in the identification of newly isolated species (56). In our study, the values of both ANI and dDDH indicated that the genome of *B. pishanensis* CHPC 1.3453$^T$ and those of taxonomically closely related *Brevundimonas* spp. showed low similarities,

implying that CHPC 1.3453[T] should be a new species. Our further experiments also showed that CHPC 1.3453[T] can be differentiated from other *Brevundimonas* species in the abilities to assimilate multiple biochemical substrates. Notably, Liu et al. used ANI and DDH data to identify 29 *Brevundimonas* taxa and found that at least 17 taxa should be assigned to novel *Brevundimonas* species (57). In this study, we did a comprehensive analysis for the *Brevundimonas* genomes. Still, two groups of *Brevundimonas* spp. (i.e., *B. diminuta* ATCC 11568[T] - *B. vancanneytii* NCTC 9239 and *B. abyssalis* TAR-001[T] - *B. denitrificans* TAR-002[T]) were found to need correction and update of the taxonomy and classification. *B. vancanneytii* NCTC 9239 should be renamed *B. diminuta* NCTC 9239. Two strains, *B. abyssalis* TAR-001[T] and *B. denitrificans* TAR-002[T], had been mislabeled, and actually they belonged to the same species, *B. abyssalis*.

## MATERIALS AND METHODS

**Strain information.** Strain CHPC 1.3453[T] was recovered from the stool sample of a 46-year-old Uygur (one of the Chinese ethnic minorities) male, living in Pishan County of Hotan Prefecture, Xinjiang, China on 27 November 2018. The sample was collected under aseptic conditions. Then, bacteria were isolated, purified, and incubated in lysogeny broth (LB) medium with 1% sodium chloride (NaCl; wt/vol) at 37°C for 24 to 48 h. In addition, 24 genomic sequences of *Brevundimonas* type strains (or representative strains) were used in this study, and the data are available in GenBank as of June 2021 (https://www.ncbi.nlm.nih.gov/genbank/, see Table S1).

**16S rRNA PCR and genome sequencing.** The DNA of strain CHPC 1.3453[T] was extracted by genomic DNA purification kit (Promega, USA) in accordance with the manufacturer's instructions. PCR amplification was conducted with bacterial universal primers (27F and 1492R) (58, 59). The 16S rRNA gene sequencing and the whole-genome sequencing were entrusted to RuiBiotech Co., Ltd. (Beijing, China) and Beijing Genomics Institute (BGI, Shenzhen, China), respectively. The genomic DNA was sequenced using Illumina HiSeq 2000 platform (Illumina Inc., San Diego, CA, USA) with a depth of 200× coverage. Library construction (paired-end reads sizes of 150 bp), genome sequencing, and data pipelining were performed by following the manufacturer's protocols.

**Genome evaluation, assembly, and annotation.** FastQC (http://www.bioinformatics.babraham.ac.uk/projects/fastqc/) was used to evaluate the raw sequence data of strain CHPC 1.3453[T], and the low-quality reads were filtered. Then, the clean data were assembled into contigs using SPAdes version 3.8.2 software (60) with default parameters. After removing contigs of <500 bp, QUAST (version 4.6.3) software (61) was applied to evaluate the genome assembly. Open reading frame (ORF) prediction and annotation were done by prodigal version 2.6.3 (62) and Prokka version 1.13.3 (63) software.

**Phylogeny analysis.** For 16S rRNA gene phylogenetic analysis, the sequence of strain CHPC 1.3453[T] and the reference sequences of *Brevundimonas* spp. available from GenBank (Table S2) were grouped together for alignment and comparison by MEGA software version 7.0.21 (64). Phylogenetic trees were constructed by both neighbor-joining (65) and maximum-likelihood (66) algorithms with 1,000 bootstrap replications. For multilocus sequence analysis (MLSA), 5 housekeeping genes (*gyrB*, *ppsA*, *recN*, *rpoC*, and *rpoD*) were selected and these gene sequences were extracted from the genomes of strain CHPC 1.3453[T] as well as 23 other type strains or representative strains from the genus *Brevundimonas*. The maximum-likelihood (ML) tree was reconstructed using PhyML version 3.1 (67) based on the above-mentioned sequences with 1,000 bootstraps.

**Comparative genomic analysis of *Brevundimonas* spp.** Orthologous gene clusters from *Brevundimonas* spp. were identified using InParanoid version 4.1 (68). The pairwise homologous gene rate (PHGR) was calculated by the distribution of orthologous gene clusters. A Perl script based on previous algorithm (22) was used to compute the average nucleotide identity (ANI). The digital DNA-DNA hybridization (dDDH) value was calculated by the GGDC web server (http://ggdc.dsmz.de/). Pan genome analysis was performed by Roary pipeline (69) with a relative relax cutoff of identity referring to the previous study (70). The analysis of multiple genome alignments in the presence of genome collinearity was done by MAUVE software version 2.4.0 (71). Genomic islands (GIs) were analyzed using online tool IslandViewer (http://www.pathogenomics.sfu.ca/islandviewer/) with default parameters. The phage/prophage sequences were predicted by PHASTER (PHAge Search Tool Enhanced Release, http://phaster.ca). The identification of CRISPR-Cas system was done by CRISPRCasFinder (https://crisprcas.i2bc.paris-saclay.fr/CrisprCasFinder/), while the detection of plasmid replicon was done by PlasmidFinder (http://www.genomicepidemiology.org/). The Venn diagram was drawn by R software (version 4.0.2).

**Virulence gene and antimicrobial resistance gene analysis.** The VFDB (Virulence Factor Database, http://www.mgc.ac.cn/VFs/) was used to predict and annotate the virulence genes. Protein sequences from the *Brevundimonas* strains were searched against the VFDB using BLASTp with identity of ≥90%, length coverage of ≥60%, and an E value of 1e−5. The antimicrobial resistance genes were detected by ResFinder tool (https://cge.cbs.dtu.dk/services/ResFinder/) and were cross-compared with the Comprehensive Antibiotic Resistance Database (CARD, http://arpcard.mcmaster.ca). The BLASTp cutoff values were set as follow: E value of 1e−5, identity of ≥90%, and length coverage of ≥60%.

**Growth curve test.** Strain CHPC 1.3453[T] as well as the type strains of *B. vesicularis* (NBRC 12165[T]), *B. diminuta* (ATCC 11568[T]), and *E. coli* (ATCC 25922[T]) were incubated in LB medium at 37°C for 3 to 4 h to the exponential phase to ensure maximum cell viability. Then, 1.25× serial dilutions were made by LB medium, and the final dilution was 100-fold. The diluted bacterial cells were spread into a 96-well plate

with a volume of 200 μL each. The changes of optical density (OD) values were detected using automated turbidity system Bioscreen C (Oy Growth Curves Ab Ltd., Raisio, Finland). Then, a growth curve was drawn with the culture time as the abscissa and the logarithm of the number of bacteria or the growth rate as the ordinate.

**Identification of biochemical phenotype.** The biochemical reactions toward various substrates were determined by API 20NE and API 50CH systems (bioMérieux), according to the manufacturers' instructions. Enzyme activities and other biochemical properties were conducted by API ZYM assay (bioMérieux). The biochemical phenotypes of strain CHPC 1.3453$^T$ were compared with those of three type trains, *B. diminuta* ATCC 11568$^T$, *B. vesicularis* NBRC 12165$^T$, and *B. halotolerans* MCS24$^T$, in each independent experiment.

**Antimicrobial susceptibility test.** The antimicrobial susceptibility testing (AST) panel for aerobic Gram-negative bacilli (Shanghai Fosun Long March Medical Science Co., Ltd., China) was used to perform the drug susceptibility test using microdilution method according to the Clinical and Laboratory Standards Institute (CLSI) guidelines (72). The bacterial cell culture of strain CHPC 1.3453$^T$ was adjusted to 0.5 McF (McFarland) and then was spread onto antibiotic plates. After an incubation period of 18 to 20 h, the results were interpreted. *E. coli* ATCC 25922$^T$ was used as a quality control. MIC was used to categorize the strain as susceptible, intermediate, or resistant.

**Data availability.** The draft genome sequence of the strain has been deposited at the NCBI GenBank under BioProject number PRJNA780817 and accession number JAJKBG000000000.

## SUPPLEMENTAL MATERIAL

Supplemental material is available online only.
**SUPPLEMENTAL FILE 1**, PDF file, 1.5 MB.

## ACKNOWLEDGMENTS

The work was supported by the National Science and Technology Infrastructure (NPRC-32) and National Science and Technology Fundamental Resources Investigation Program of China (2021FY100900). We have no conflicts of interest.

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
