## [Reviewer comments · Microbiology Spectrum]

Microbiology Spectrum

Comparative Genomic Analysis Reveals Potential Pathogenicity and Slow-growth Characteristics of genus *Brevundimonas* and Description of *Brevundimonas pishanensis* sp. nov.

Zhenzhou Huang, Keyi Yu, Yue Xiao, Yonglu Wang, Di Xiao, and Duochun Wang

Corresponding Author(s): Duochun Wang, National Institute for Communicable Disease Control & Prevention, China CDC

Review Timeline:

Submission Date:	December 2, 2021
Editorial Decision:	March 12, 2022
Revision Received:	March 21, 2022
Accepted:	March 26, 2022

Editor: Daria Van Tyne

Reviewer(s): Disclosure of reviewer identity is with reference to reviewer comments included in decision letter(s). The following individuals involved in review of your submission have agreed to reveal their identity: Rhys T White (Reviewer #2)

Transaction Report:

DOI: <https://doi.org/10.1128/spectrum.02468-21>

March 12, 2022

Prof. Duochun Wang
National Institute for Communicable Disease Control & Prevention, China CDC
Changbai Road 155, Changping
Beijing 102206
China

Re: Spectrum02468-21 (Comparative Genomic Analysis Reveals Potential Pathogenicity and Slow-growth Characteristics of genus *Brevundimonas* and Description of *Brevundimonas pishanensis* sp. nov.)

Dear Prof. Duochun Wang:

Thank you for submitting your manuscript to Microbiology Spectrum and apologies for the long delay, it was unusually difficult to find reviewers for your submission. Your manuscript has been reviewed by two experts, and I would now like you to modify your study in line with their comments below.

Link Not Available

Sincerely,

Daria Van Tyne

Journals Department
Reviewer comments:

Reviewer #1 (Comments for the Author):

The authors performed comparative genomic analysis in 24 *Brevundimonas* species with a novel species they isolated in China. Phylogenetic analysis was performed using multiple approaches. Virulence/antibiotic associated genes were predicted using bioinformatics tools. Collectively, the evidence support that the newly isolated strain belongs to a novel species.

Major comments:

1. The authors grouped the different species into 5 "clades" using hierarchical clustering method based on pan genome diversity

which was based on gene presence and absence profile. However, the data seems to be contradictory to that from ANI and core genome tree. According to the core genome tree in Fig2A, TAR001 and TAR002 should not have been grouped into one same clade. Additionally, depending on the cutoff being used, the "clade" grouping could end up differently. The authors need to define the cutoff in the main text.

2. line 232: "... had a highly similar topology...": the reviewer disagrees with the conclusion. The phylogenetic tree from Fig 2A and Fig 2B differs at the clustering of TAR001/TAR002 and *B. lutea* NS26/*B. aveniformis* DSM17977.

3. Characterization of antibiotic resistance profile requires data support, such as MIC, which is missing in the main text. As for "slow growth", it is recommended the authors provide the doubling time.

Minor comments:

1. line 62: "tiny": it's better to use size in micro-meter instead of a generic "tiny".

2. line 66: "human beings": please specify location, eg on skin or colonized internally or opportunistic pathogen.

3. line 82: "few articles": please cite the related references.

4. line 219: "5 major clades": what's the cutoff value? what's the rational of using Fig 1C to define clades instead of Fig1B, 2A or 2B?

5. line 248: "data not shown": recommend to add this data to supplemental material.

6. line 291-304: why were different strains used in Fig 4A and 4B?

7. line 496: "two groups of ...": it is not clear which two groups are being discussed here. Was this data in the main text?

Reviewer #2 (Comments for the Author):

Really great work! I enjoyed reading this comprehensive manuscript. I have a couple of comments that I would like to be addressed.

Major:

1) Line 19: "Brevundimonas spp. have relatively small genomes (3.13{plus minus}0.29 Mb)". Please note, bacteria can have a genome size between ~0.5 to ~14 Mbp. Please can you reword this as 3.13 Mbp is not relatively small when compared to other bacteria like *Chlamydia* spp. (~1.1Mbp). Also, could you please define what the +/- is representing? Standard deviation? Standard error of mean?

2) Line 109: Please change "The genome of strain CHPC 1.3453T was extracted" to "The DNA of strain CHPC 1.3453T was extracted"

3) Line 126: Was a reference used for QUAST? If so which reference was used?

4) Line 132: What alignment was performed? Muscle? ClustralW?

Figure 4: What does the axis represent? Please describe this is figure legend

Figure S2: What does the scale bar represent in the tree? How is the tree rooted? Midpoint? Outgroup? What do the numbers on the nodes represent? Bootstrap? How many replicates? When I refer back to the methods, I can see that 1,000 bootstrap replicated was done. Might be worth while to state this in the figure legend for readability.

Figure S4B: What does the scale bar represent in the tree? What does the axis represent? How is the tree rooted? Midpoint? Outgroup? What do the black dots on the nodes represent? What do the numbers on the nodes represent? Bootstrap? How many replicates? When I refer back to the methods, I can see that 1,000 bootstrap replicated was done. Might be worth while to state this in the figure legend for readability.

Minor:

1) Only a minor point, so please feel free to disregard this comment. I can appreciate that English may not be everyones first language. Maybe the authors can revisit if they would like to use American (USA) English spellings or British English spellings. E.g., Faeces (British) is spelt as feces (USA) above. Bacteraemia (British) or Bacteremia (USA), diarrhoea (British) or diarrhea (USA), favourable (British) or favorable (USA)... ect.

2) Line 22: phylogenetic is spelt incorrectly

3) Line 32: Please change "a diarrhea patient" to "a patient(suffering with diarrhea)"

4) Line 34: Please define what MLSA is an abbreviation of and replace "as well as" with "and"

5) Line 68: Please change "human beings" to "human hosts"

6) Line 102: Please change "46-year-old Uygur (one of the Chinese ethnic minorities) man" to "46-year-old Uygur (one of the Chinese ethnic minorities) male"

7) Line 105: Quick note: I believe that LB stands for "Lysogeny broth", not Luria-Bertani medium- or variations of that. I could be wrong, but I am basing this comment from the original 1951 reference. Please note, Luria-Bertani is also described in the legend to Figure S5

8) Line 107: Data availability. Please specify what data is available in GenBank The sequence read data? Or the assemblies? Please note, sequence read data is stored on the sequence read archive (SRA), not GenBank (Assemblies). Can you please refer to the BioProject number?

9) Line 114: Change "The genome DNA" to "The genomic DNA"

10) Line 145: Please define abbreviations when used first. I.e., digital DNA-DNA hybridisation (dDDH)

11) Line 259: Please change "with VFDB" to "with the VFDB"

12) Line 629: Please change "the gene *kdsA* and *acpXL* were detected in 95.8% (23/24)" to "the gene *kdsA* and *acpXL* were detected in 95.8% (n = 23/24)"

13) Line 272: Please change "The majority of the *Brevundimonas* species (87.5%, 21/24)" to "The majority of the *Brevundimonas* species (87.5%, n = 21/24)"

14) Line 294: What kind of alignment?

Staff Comments:

Preparing Revision Guidelines

Please return the manuscript within 60 days; if you cannot complete the modification within this time period, please contact me. If you do not wish to modify the manuscript and prefer to submit it to another journal, please notify me of your decision immediately so that the manuscript may be formally withdrawn from consideration by Microbiology Spectrum.

**Comparative Genomic Analysis Reveals Potential Pathogenicity and**
**Slow-growth Characteristics of genus *Brevundimonas* and Description of**
***Brevundimonas pishanensis* sp. nov.**

Zhenzhou Huang^{a,b}, Keyi Yu^{a,b}, Yue Xiao^{a,b}, Yonglu Wang^c, Di Xiao^a, Duochun
Wang^{a,b*}

6 ^a National Institute for Communicable Disease Control and Prevention, Chinese
Center for Disease Control and Prevention (China CDC), State Key Laboratory of
Infectious Disease Prevention and Control, Beijing, 102206, PR China.

9 ^b Center for Human Pathogenic Culture Collection, China CDC, Beijing, 102206, PR
China.

11 ^c Ma'anshan Center for Disease Control and Prevention, Anhui Province, PR China.

*Correspondence: Duochun Wang, wangduochun@icdc.cn. Changbai Road 155,
Changping, Beijing, 102206, PR China.

**ABSTRACT** The genus *Brevundimonas* consists of Gram-negative bacteria
widely distributed in environment and can cause human infections. However, the
genomic characteristics and pathogenicity of *Brevundimonas* remain poorly studied.
Here, the whole-genome features of 24 *Brevundimonas* type strains were described.
*Brevundimonas* spp. have relatively small genomes (3.13±0.29 Mb) but high G+C
contents (67.01±2.19 mol%). Two-dimensional hierarchical clustering divided those
genomes into 5 major clades, in which *Clade II* and *V* contained nine and five species,
respectively. Interestingly, phylogenetic analysis showed a one-to-one match between
core- and accessory-genome, which suggests co-evolution of species within the genus
*Brevundimonas*. The unique genes were annotated to biological functions like
catalytic activity, translation and multi-substance metabolism, *et al.* The majority of
*Brevundimonas* spp. harbored virulence-associated genes *icl*, *tufA*, *kdsA*, *htpB* and
*acpXL*, which encoded isocitrate lyase, elongation factor,
2-dehydro-3-deoxyphosphooctonate aldolase, heat shock protein and acyl carrier
protein, respectively. In addition, genomic islands (GIs) and phages/prophages were

identified within the *Brevundimonas* genus. Importantly, a novel *Brevundimonas*
species was identified from the feces of a patient (suffering with diarrhea) by the
analyses of biochemical characteristics, phylogenetic tree of 16S rRNA gene and
Multilocus sequence analysis (MLSA) sequences, and genomic data. The name
*Brevundimonas pishanensis* sp. nov. was proposed, with type strain CHPC 1.3453^T
(=GDMCC 1.2503^T= KCTC 82824^T). *Brevundimonas* spp. also showed obvious slow
growth compared with *Escherichia coli*. Our study reveals insights into genomic
characteristics and potential virulence-associated genes of *Brevundimonas* spp., and
provides a basis for further intensive study of the pathogenicity of *Brevundimonas*.

**IMPORTANCE** *Brevundimonas* spp., a group of bacteria from the family
*Caulobacteraceae* and associated with nosocomial infections, deserve widespread
attention. Our study elucidated genes potentially associated with the pathogenicity of
the *Brevundimonas* genus. We also described some new characteristics of
*Brevundimonas* spp., such as small chromosome size, high G+C content and
slow-growth phenotypes, which made the *Brevundimonas* genus as a good model
organism for in-depth studies of growth rate traits. Apart from the comparative
analysis of the genomic features of the *Brevundimonas* genus, we also reported a
novel *Brevundimonas* species, *Brevundimonas pishanensis*, from the feces of a patient
with diarrhea. Our study promotes the understanding of the pathogenicity
characteristics of *Brevundimonas* spp. bacteria.

**KEYWORDS** comparative genomics; *Brevundimonas*; pathogenicity; slow-growth;
new species proposed.

**INTRODUCTION**

Gram-negative, non-fermenting bacteria have raised increasing concern in clinical
practice, since they are one of the most common causes of nosocomial infection.
Among these, some are well known opportunistic pathogens associated with
hospital-acquired infections, for example, *Pseudomonas aeruginosa* (1),
*Acinetobacter baumannii* (2, 3), and *Enterococcus faecium* (4). *Brevundimonas* spp.
are relatively less known, but they are also opportunistic human pathogens potentially

related to hospital infections.

The genus *Brevundimonas*, first described by Segers et al. (5) in 1994, comprises a
group of bacteria shared the basic microbiological characteristics, like Gram-negative,
motile, tiny, non-fermenting, oxidase-positive, and aerobic or facultative anaerobic.
There are currently 35 species within this genus
(<https://www.bacterio.net/genus/brevundimonas>). A profusion of new members can be
isolated from diverse sources, such as soil (6-9), lake or sea sediment (10-12),
activated sludge (13), aquatic water (14, 15) and human hosts (16-19). In humans,
*Brevundimonas* spp. were rarely isolated and occasionally caused disease in patients
with impaired immunity (20). The most common and clinically relevant pathogenic
species in the *Brevundimonas* genus are *B. vesicularis* (related to approximately 70%
cases reported) and *B. diminuta* (related to more than 20% cases reported). Besides,

[revised manuscript text omitted]

Cells of strain CHPC 1.3453^T were motile by a single polar flagellum. The strain
formed circular, orange-yellow colonies with a diameter of 0.5-1.5 mm on LB agar
and grey colonies on Blood agar at 35°C for 18-72 h, with translucent texture, round
shape, and plump appearance (Fig. 5A, 5B). Strain CHPC 1.3453^T was small, short,
rod-shaped, under high-resolution transmission electron microscopy (Fig. 5C). Cell

size was 1.2-4 μm in length and 0.4 μm in width. Phenotypic results showed that
strain CHPC 1.3453^T was Gram-stain-negative, oxidase-positive; it can grow on LB
medium, Blood medium, BHI medium, or R2A agar; it was able to grow under a wide
range of pH (pH 6-10, optimum pH 8) and NaCl tolerance concentrations (0-3%, w/v);
it can grow at a temperature range of 15-42°C (it did not grow at 4°C); the optimum
growth temperature was recommended to be 30-37°C; it was facultative aerobic. The
strain was susceptible to amikacin, amoxicillin-clavulanate, ampicillin-sulbactam,
cefepime, cefoperazone-sulbactam, cefoxitin, ceftazidime, ceftriaxone, cefuroxime,
chloramphenicol, ciprofloxacin, colistin, ertapenem, fosfomicin, gentamicin,
imipenem, levofloxacin, meropenem, minocycline, moxifloxacin, norfloxacin,
piperacillin-tazobactam, tetracycline, tigecycline, and tobramycin but resistant to
aztreonam, ceftazolin, nitrofurantoin, and trimethoprim-sulfamethoxazole.

[revised manuscript text omitted]

1.3453^T. **(A)** The core- and accessory- genome phylogeny of 24 *Brevundimonas* spp.
strains. The colors of each species correspond to the colors of five different clades in
the Fig. 1C. **(B)** The MAUVE comparison of CHPC 1.3453^T and closely related
members in the *Clade V* of pan genome clustering in Fig. 1C. **(C)** The Venn diagram
of the shared and unique genes found in the novel strain CHPC 1.3453^T and other
closely related members in *Clade V*.

**FIG 3** The distribution of virulence-associated genes of *Brevundimonas* spp.. **(A)**
Hierarchically clustering heatmap of 24 *Brevundimonas* spp. with respect to the
virulence gene sources. The colors from blue to red represent the correlation
frequency of virulence gene homology. **(B)** Hierarchically clustering heatmap of
virulence-associated genes in 24 *Brevundimonas* spp. The colors represent the
sequences-BLAST coverage values of virulence-associated genes.

**FIG 4** Phylogenetic trees of *Brevundimonas* species. **(A)** The neighbour-joining tree
based on 16S rRNA gene sequences of the strain CHPC 1.3453^T and other
*Brevundimonas* spp. isolates. Bootstrap values below 70%, based on 1000
re-samplings, are not shown at branch nodes. Filled black circles at nodes indicate
generic branches that are synchronously recovered by using neighbour-joining and
maximum-likelihood algorithms. GenBank accession numbers of the 16S rRNA gene
sequences are given in parentheses. Three strains, namely *Caulobacter fusiformis*
ATCC 15257^T, *Asticcacaulis excentricus* DSM 4724^T, and *Sphingomonas adhaesiva*
DSM 7418^T, respectively, are served as outgroups. The horizontal bar represents 0.01
substitution per nucleotide site. **(B)** The maximum-likelihood tree based on 5
housekeeping gene concatenated sequences, in the order of

*gyrB-ppsA-recN-rpoC-rpoD*. Numbers at nodes indicate bootstrap values (percentage
of 1000 replicates) greater than 70%. Filled black circles indicate generic branches
that are also recovered by using neighbour-joining and maximum-likelihood
algorithms. The horizontal bar represents 0.1 substitution per nucleotide site.

**FIG 5** Culture and morphology characteristics of strain CHPC 1.3453^T. **(A)** The
growth of this strain after 18 h at 35°C on Blood Agar medium. **(B)** The morphology
of single colony after 72 h growth at 35°C on Luria-Bertani (LB) medium. The bar
represents a unit length of 10 millimeters. **(C)** The general morphology of a
negatively-stained bacterial cell showed by high resolution transmission electron. The
562 bar represents a unit length of 1000 nanometers. **(D)** Growth curves of strain CHPC
1.3453^T. Three strains, *B. vesicularis* NBRC 12165^T and *B. diminuta* ATCC 11568^T
and *E. coli* ATCC 25922^T, were used controls.

REFERENCES

[revised manuscript text omitted]

Authors: Zhenzhou Huang, Keyi Yu, Yue Xiao, Yonglu Wang, Di Xiao, and Duochun Wang

Reviewer #1 (Comments for the Author):

The authors performed comparative genomic analysis in 24 *Brevundimonas* species with a novel species they isolated in China. Phylogenetic analysis was performed using multiple approaches. Virulence/antibiotic associated genes were predicted using bioinformatics tools. Collectively, the evidence support that the newly isolated strain belongs to a novel species.

Response:

Dear reviewer, we would like to thank you for your careful reading, helpful comments. We have carefully considered all comments and revised our manuscript accordingly.

Major comments:

1. The authors grouped the different species into 5 "clades" using hierarchical clustering method based on pan genome diversity which was based on gene presence and absence profile. However, the data seems to be contradictory to that from ANI and core genome tree. According to the core genome tree in Fig2A, TAR001 and TAR002 should not have been grouped into one same clade. Additionally, depending on the cutoff being used, the "clade" grouping could end up differently. The authors need to define the cutoff in the main text.

Response:

Thank you very much for your constructive suggestions. Defining clade in this study was performed using two approaches, including pan-genome variation and core-genome Maximum-likelihood method. The above methods get a consistent clade division, although they each have a corresponding cutoff value. Specifically, pan-genome analysis has a cutoff (based gene-presence/absence profile similarity) as 25.6%, while core-genome tree has a SNP-threshold of 500 SNPs for defining *Brevundimonas* genus into 5 clades. In core genome tree in Fig2A, it seems a bit misleading, because both the core- and accessory- genome trees only use the "Rectangular-Display only topology" visualization mode in the software, without the branch length. In fact, all the data (pan genome diversity, ANI, core- and accessory- genome phylogeny) consistently show that TAR001 and TAR002 are very closely related and should be grouped together. We have replaced the evolutionary trees (with branch length) in Figure 2A, and also marked the cutoff value of 'clade division' in the new manuscript.

2. line 232: "... had a highly similar topology...": the reviewer disagrees with the conclusion. The phylogenetic tree from Fig 2A and Fig 2B differs at the clustering of TAR001/TAR002 and *B. lutea* NS26/*B. aveniformis* DSM17977.

Response:

Thank you very much for your comments. The topology in Figure 2A seems to be a bit unclear, so that the reviewers have raised questions. To solve this problem, we changed the way the evolutionary tree displayed in the new Figure 2A and added a cutoff criterion for clade division.

3. Characterization of antibiotic resistance profile requires data support, such as MIC, which is missing in the main text. As for "slow growth", it is recommended the authors provide the doubling time.

Response:

Thanks very much for reviewer's comments. We have added the MIC value to the results of antibiotic resistance test (Line 338-349). We also provide the doubling time of bacterial growth (*B. pishanensis* sp. nov. = 75 min; *B. diminuta* = 81 min; *B. vesicularis* = 135 min; *E. coli*=27 min) in the new manuscript Line 360-364.

Minor comments:

1. line 62: "tiny": it's better to use size in micro-meter instead of a generic "tiny".

Response:

Thanks very much for pointing out the inappropriateness. We have made a correction in the new manuscript Line 64.

2. line 66: "human beings": please specify location, eg on skin or colonized internally or opportunistic pathogen.

Response:

Thanks very much for reviewer's comments. We have made a revision in the new manuscript Line 69.

3. line 82: "few articles": please cite the related references.

Response:

Thanks very much for reviewer's comments. we have cited the related references in the new manuscript in the Line 84.

References as follow:

16. Yang ML, Chen YH, Chen TC, Lin WR, Lin CY, Lu PL. 2006. Case report: infective endocarditis caused by *Brevundimonas vesicularis*. *BMC INFECT DIS* 6:179.

17. Zhang CC, Hsu HJ, Li CM. 2012. *Brevundimonas vesicularis* bacteremia resistant to trimethoprim-sulfamethoxazole and ceftazidime in a tertiary hospital in southern Taiwan. *J Microbiol Immunol Infect* 45:448-52.

18. Lee MR, Huang YT, Liao CH, Chuang TY, Lin CK, Lee SW, Lai CC, Yu CJ, Hsueh PR. 2011. Bacteremia caused by *Brevundimonas* species at a tertiary care hospital in Taiwan, 2000-2010. *Eur J Clin Microbiol Infect Dis* 30:1185-1191.

19. Gilad J, Borer A, Peled N, Riesenber K, Tager S, Appelbaum A, Schlaeffer F. 2000. Hospital-acquired *brevundimonas vesicularis* septicaemia following open-heart surgery: case report and literature review. *Scand J Infect Dis* 32:90-91."

4. line 219: "5 major clades": what's the cutoff value? what's the rational of using Fig 1C to define clades instead of Fig1B, 2A or 2B?

Response:

Thanks very much for reviewer's comments. We have added a cutoff criterion for clade division in the new manuscript Line 225-226. Defining clade in this study was performed both pan-genome diversity, and core-genome Maximum-likelihood method. The above methods get a consistent clade division, although they each have a corresponding cutoff value. The clustering dendrogram was done by R (*hclust function*) using 'complete' parameter. Using pan-genome variation to divide clades, because pan-genome analysis not only includes information about the gene presence and absence profile of core genes, but also reflects the information of accessory-genes. It took into account the evolutionary differences of common sites, and also proved that the existence of some unique gene segments, which may be related to horizontal gene transfer. We also refer many recent publications^[1-4] about "defining clade" based on pan-genome variation.

1. Kim Y, Gu C, Kim HU, Lee SY. Current status of pan-genome analysis for pathogenic bacteria. *Curr Opin Biotechnol*. 2020 Jun;63:54-62. doi: 10.1016/j.copbio.2019.12.001. Epub 2019 Dec 28. PMID: 31891864.
2. Oshkin IY, Miroshnikov KK, Grouzdev DS, Dedysh SN. Pan-Genome-Based Analysis as a Framework for Demarcating Two Closely Related Methanotroph Genera *Methylocystis* and *Methylosinus*. *Microorganisms*. 2020 May 20;8(5):768. doi: 10.3390/microorganisms8050768. PMID: 32443820; PMCID: PMC7285482.
3. Zhou Z, Lundstrøm I, Tran-Dien A, Duchêne S, Alikhan NF, Sergeant MJ, Langridge G, Fotakis AK, Nair S, Stenøien HK, Hamre SS, Casjens S, Christophersen A, Quince C, Thomson NR, Weill FX, Ho SYW, Gilbert MTP, Achtman M. Pan-genome Analysis of Ancient and Modern *Salmonella enterica* Demonstrates Genomic Stability of the Invasive Para C Lineage for Millennia. *Curr Biol*. 2018 Aug 6;28(15):2420-2428.e10. doi: 10.1016/j.cub.2018.05.058. Epub 2018 Jul 19. PMID: 30033331; PMCID: PMC6089836.
4. Brüggemann H, Jensen A, Nazipi S, Aslan H, Meyer RL, Poehlein A, Brzuszkiewicz E, Al-Zeer MA, Brinkmann V, Söderquist B. Pan-genome analysis of the genus *Finegoldia* identifies two distinct clades, strain-specific heterogeneity, and putative virulence factors. *Sci Rep*. 2018 Jan 10;8(1):266. doi: 10.1038/s41598-017-18661-8. PMID: 29321635; PMCID: PMC5762925.

5. line 248: "data not shown": recommend to add this data to supplemental material.

Response:

Thanks very much for reviewer's comments. We have added this part of the data (GO, COG, KEGG annotation in the Fig. S4) and reword in the new manuscript Line 257-260.

6. line 291-304: why were different strains used in Fig 4A and 4B?

Response:

Thanks very much for reviewer's comments. Fig 4A was done using all the 16S

rRNA gene sequences currently available on the GenBank. As a result, there are 33 sequences. As for the MLSA analysis in the Fig 4B, the sequence of each housekeeping gene was extracted from the whole genome sequence. As a result, there are currently only 24 (including the new species in this study) genome sequences.

7. line 496: "two groups of ...": it is not clear which two groups are being discussed here. Was this data in the main text?

Response:

Thanks very much for reviewer's comments. Line 208-220: "*The overall ANI values between any two representative genomes, were under the classical boundary of 95% - 96% (33,38) for an independent species or subspecies (Fig. 1B), except for two groups, i.e., B. diminuta ATCC 11568T - B. vancouverii NCTC 9239, and B. abyssalis TAR-001T - B. denitrificans TAR-002T. It was suggested that each group belongs to synonyms.*"

In order to avoid confusion for readers, we have re-declared the 'two groups' that appeared in the Discussion section Line 521-524.

Reviewer #2 (Comments for the Author):

Really great work! I enjoyed reading this comprehensive manuscript. I have a couple of comments that I would like to be addressed.

Response:

Thanks very much for reviewer's careful review and helpful suggestions. We have made point-by-point responses to the issues raised by the reviewers in the new manuscript.

Major:

1) Line 19: "Brevundimonas spp. have relatively small genomes (3.13{plus minus}0.29 Mb)". Please note, bacteria can have a genome size between ~0.5 to ~14 Mbp. Please can you reword this as 3.13 Mbp is not relatively small when compared to other bacteria like Chlamydia spp. (~1.1Mbp). Also, could you please define what the +/- is representing? Standard deviation? Standard error of mean?

Response:

Thanks very much for reviewer's comments. Firstly, our data following the normal distribution, '+/-' is 'Mean \pm Standard deviation'. As the reviewer pointed out, bacteria can have a genome size between ~0.5 to ~14 Mbp. Brevundimonas spp. was subordinate to Family Caulobacteraceae, Order Caulobacterales. We have

done a statistics for the size of the Family Caulobacteraceae bacterial genome larger to be 4.49 ± 0.83 Mb (*t* test, $P < 0.001$). Therefore, we determined that the bacterial genome size of *Brevundimonas* spp. is relatively small in the whole Family Caulobacteraceae bacteria. We have made a supplementary note in the Line 19 and 'Result' section of the new manuscript Line 196-198.

2) Line 109: Please change "The genome of strain CHPC 1.3453T was extracted" to "The DNA of strain CHPC 1.3453T was extracted"

Response:

Thanks very much for reviewer's comments. We have changed 'the genome' to 'the DNA' in the new manuscript Line 112.

3) Line 126: Was a reference used for QUASt? If so which reference was used?

Response:

Thanks very much for reviewer's comments. The published paper, which citation format is "Gurevich A, Saveliev V, Vyahhi N, Tesler G. 2013. QUASt: quality assessment tool for genome assemblies. *BIOINFORMATICS* 29:1072-1075" has been cited as Reference No.25 in the new manuscript Line 127.

4) Line 132: What alignment was performed? Muscle? ClustralW?

Response:

Thanks very much for reviewer's comments. The alignment of core genome sequences was aligned by Roary software automatically; while the alignment of 16S rRNA gene and multilocus sequence analysis (MLSA), we used the 'ClustralW module' that comes with the MEGA software version 7.0.21.

Figure 4: What does the axis represent? Please describe this is figure legend

Response:

Thanks very much for pointing out the inappropriateness. The axis represents the length of evolutionary branches. At present, we have used the horizontal bar as a tree scale in the figure; and to avoid redundancy, we have removed the axis in the new version of the Figure 4.

Figure S2B: What does the scale bar represent in the tree? How is the tree rooted? Midpoint? Outgroup? What do the numbers on the nodes represent? Bootstrap? How many replicates? When I refer back to the methods, I can see that 1,000 bootstrap replicated was done. Might be worth while to state this in the figure legend for readability.

Response:

Thanks very much for reviewer's comments. In the Figure S2, they are not the scale bars. The true scale is located at the bottom of this image. Those colored horizontal lines represent different genus in the Family Caulobacteraceae. Those non-*Brevundimonas* genus genomes served as outgroups in the phylogenetic tree, which helps to root the tree and helps to locate the evolutionary position of *Brevundimonas* genus in the Family Caulobacteraceae. Bootstrap values over 70%,

based on 1000 resamplings, are shown at branch nodes. We have revised it in the 'Materials and Methods' section and also marked in the figure legend of Figure S2.

Figure S4B: What does the scale bar represent in the tree? What does the axis represent? How is the tree rooted? Midpoint? Outgroup? What do the black dots on the nodes represent? What do the numbers on the nodes represent? Bootstrap? How many replicates? When I refer back to the methods, I can see that 1,000 bootstrap replicated was done. Might be worth while to state this in the figure legend for readability.

Response:

Thanks very much for reviewer's comments. This question is a duplicate of the above, and we have revised it according to the reviewers' good suggestions. We are very grateful to the reviewer.

Minor:

1) Only a minor point, so please feel free to disregard this comments. I can appreciate that English may not be everyones first language. Maybe the authors can revisit if they would like to use American (USA) English spellings or British English spellings. E.g., Faeces (British) is spelt as feces (USA) above. Bacteraemia (British) or Bacteremia (USA), diarrhoea (British) or diarrhea (USA), favourable (British) or favorable (USA)... ect.

Response:

Thanks very much for reviewer's suggestions for revisions in the language. We carefully revisited the wording habits in the manuscript, and switched to the American English language style uniformly.

2) Line 22: phylogenetic is spelt incorrectly

Response:

Thanks very much for pointing out the inappropriateness. We have corrected the spelling of this word in the new manuscript Line 23.

3) Line 32: Please change "a diarrhea patient" to "a patient(suffering with diarrhea)"

Response:

Thanks very much for reviewer's suggestions. We changed to "suffering from diarrhea" in the Line 32 and other place throughout new manuscript.

4) Line 34: Please define what MLSA is an abbreviation of and replace "as well as" with "and"

Response:

Thanks very much for reviewer's comments. MLSA is the abbreviation of 'multilocus sequence analysis'; For the first occurrence of abbreviations, we define all their full name throughout the new manuscript. According to the reviewer's suggestion, we replace "as well as" with "and" in the manuscript Line 34.

5) Line 68: Please change "human beings" to "human hosts"

Response:

Thanks very much for reviewer's suggestions. We have made a revision in the new manuscript Line 69.

6) Line 102: Please change "46-year-old Uyгур (one of the Chinese ethnic minorities) man" to "46-year-old Uyгур (one of the Chinese ethnic minorities) male"

Response:

Thanks very much for reviewer's suggestions. We have made a revision in the new manuscript Line 104.

7) Line 105: Quick note: I believe that LB stands for "Lysogeny broth", not Luria-Bertani medium- or variations of that. I could be wrong, but I am basing this comments from the original 1951 reference. Please note, Luria-Bertani is also described in the legend to Figure S5

Response:

Thanks very much for reviewer's good suggestions. Lysogeny broth (LB), a nutritionally rich medium, is primarily used for the growth of bacteria. It is also known as Luria broth or Luria-Bertani broth or Lennox broth. Though the name 'Luria-Bertani broth' is very widely used. The acronym 'LB' has been variously interpreted, perhaps flatteringly, but incorrectly, as Luria broth, Lennox broth, or Luria Bertani medium. For the historical record, the abbreviation 'LB' was intended to stand for "lysogeny broth". Therefore, we have made a correction in the new manuscript Line 107-108, in the legend to Figure S5 Line 586.

References

1. Anderson, E. H. (1946). Growth requirement of virus-resistant mutants of Escherichia coli strain B. Proc. Natl. Acad. Sci. USA 32:120-128. PMID 16588724
2. Bertani, G. (1951). Studies on lysogenesis. I. The mode of phage liberation by lysogenic Escherichia coli. J. Bacteriol. 62:293-300. PMID 14888646
3. Luria, S. E., and J. W. Burrous. (1957). Hybridization between Escherichia coli and Shigella. J. Bacteriol. 74:461-476. PMID 13475269
4. Lennox, E. S. (1955). Transduction of linked genetic characters of the host by bacteriophage P1. Virology. 1:190-206. PMID 13267987
5. Luria, S. E., J. N. Adams, and R. C. Ting. (1960). Transduction of lactose-utilizing ability among strain of E. coli and S. dysenteriae and the properties of the transducing phage particles. Virology. 12:348-390. PMID 13764402
6. Miller, J. H. (1972). Experiment in molecular genetics. Cold Spring Harbor Laboratory, Cold Spring Harbor, New York.
7. Sambrook, J., E. F. Fritsch, and T. Maniatis. (1989). Molecular cloning: a laboratory manual, 2nd edition. Cold Spring Harbor Laboratory, Cold Spring Harbor, New York.
8. Bertani, G. (2004). Lysogeny at mid-twentieth century: P1, P2, and other experimental systems. J. Bacteriology. 186:595-600. PMID 14729683 doi:10.1128/JB.186.3.595-600.2004

8) Line 107: Data availability. Please specify what data is available in GenBank The sequence read data? Or the assemblies?

Please note, sequence read data is stored on the sequence read archive (SRA), not GenBank (Assemblies). Can you please refer to the BioProject number?

Response:

Thanks very much for reviewer's comments. We have uploaded the assembled genome sequence of novel species (*Brevundimonas pishanensis* sp. nov.) to NCBI (Assembly database) and obtained the BioProject number PRJNA780817 and Assembly number JAJKBG000000000. We also marked both numbers in the manuscript (line 393).

9) Line 114: Change "The genome DNA" to "The genomic DNA"

Response:

Thanks very much for reviewer's suggestions. We have made a revision in the new manuscript Line 112.

10) Line 145: Please define abbreviations when used first. I.e., digital DNA-DNA hybridization (dDDH)

Response:

Thanks very much for reviewer's suggestions. For the first occurrence of abbreviations, we define all their full name throughout the new manuscript.

11) Line 259: Please change "with VFDB" to "with the VFDB"

Response:

Thanks very much for reviewer's suggestions. We have made a revision in the new manuscript Line 266.

12) Line 629: Please change "the gene *kdsA* and *acpXL* were detected in 95.8% (23/24)" to "the gene *kdsA* and *acpXL* were detected in 95.8% (n = 23/24)"

Response:

Thanks very much for reviewer's comments. We have made a revision in the new manuscript Line 277.

13) Line 272: Please change "The majority of the *Brevundimonas* species (87.5%, 21/24)" to "The majority of the *Brevundimonas* species (87.5%, n = 21/24)"

Response:

Thanks very much for reviewer's comments. We have made a revision in the new manuscript Line 279.

14) Line 294: What kind of alignment?

Response:

Thanks very much for reviewer's comments. For phylogenetic analysis based on

16S rRNA gene sequences and MLSA concatenated sequence (in the order of gyrB-ppsA-recN-rpoC-rpoD), we used the 'ClustalW module' that comes with the MEGA software version 7.0.21 to align multiple gene sequences. On this basis, we then built an evolutionary tree by MEGA software.

March 26, 2022

Prof. Duochun Wang
National Institute for Communicable Disease Control & Prevention, China CDC
Changbai Road 155, Changping
Beijing 102206
China

Re: Spectrum02468-21R1 (Comparative Genomic Analysis Reveals Potential Pathogenicity and Slow-growth Characteristics of genus *Brevundimonas* and Description of *Brevundimonas pishanensis* sp. nov.)

Dear Prof. Duochun Wang:

Your manuscript has been accepted, and I am forwarding it to the ASM Journals Department for publication. You will be notified when your proofs are ready to be viewed.

Sincerely,

Daria Van Tyne
Editor, Microbiology Spectrum